# The Involvement of Polyamines Catabolism in the Crosstalk between Neurons and Astrocytes in Neurodegeneration

**DOI:** 10.3390/biomedicines10071756

**Published:** 2022-07-21

**Authors:** Manuela Cervelli, Monica Averna, Laura Vergani, Marco Pedrazzi, Sarah Amato, Cristian Fiorucci, Marianna Nicoletta Rossi, Guido Maura, Paolo Mariottini, Chiara Cervetto, Manuela Marcoli

**Affiliations:** 1Department of Science, University of Rome “Roma Tre”, Viale Marconi 446, 00146 Rome, Italy; cristian.fiorucci@uniroma3.it (C.F.); mariannanicoletta.rossi@uniroma3.it (M.N.R.); paolo.mariottini@uniroma3.it (P.M.); 2Neurodevelopment, Neurogenetics and Molecular Neurobiology Unit, IRCCS Fondazione Santa Lucia, Via del Fosso di Fiorano 64, 00143 Rome, Italy; 3Department of Experimental Medicine, Section of Biochemistry, University of Genova, Viale Benedetto XV 1, 16132 Genoa, Italy; monica.averna@unige.it (M.A.); marco.pedrazzi@unige.it (M.P.); 4Department of Earth, Environment and Life Sciences (DISTAV), University of Genova, Corso Europa 26, 16132 Genoa, Italy; laura.vergani@unige.it; 5Department of Pharmacy, Section of Pharmacology and Toxicology, University of Genova, Viale Cembrano 4, 16148 Genoa, Italy; amato@difar.unige.it (S.A.); maura@difar.unige.it (G.M.); 6Interuniversity Center for the Promotion of the 3Rs Principles in Teaching and Research (Centro 3R), 56122 Pisa, Italy; 7Centre of Excellence for Biomedical Research CEBR, University of Genova, Viale Benedetto XV 9, 16132 Genoa, Italy

**Keywords:** mouse genetic model, neuron damage, polyamine, reactive astrocytosis, spermine oxidase

## Abstract

In mammalian cells, the content of polyamines is tightly regulated. Polyamines, including spermine, spermidine and putrescine, are involved in many cellular processes. Spermine oxidase specifically oxidizes spermine, and its deregulated activity has been reported to be linked to brain pathologies involving neuron damage. Spermine is a neuromodulator of a number of ionotropic glutamate receptors and types of ion channels. In this respect, the Dach-SMOX mouse model overexpressing spermine oxidase in the neocortex neurons was revealed to be a model of chronic oxidative stress, excitotoxicity and neuronal damage. Reactive astrocytosis, chronic oxidative and excitotoxic stress, neuron loss and the susceptibility to seizure in the Dach-SMOX are discussed here. This genetic model would help researchers understand the linkage between polyamine dysregulation and neurodegeneration and unveil the roles of polyamines in the crosstalk between astrocytes and neurons in neuroprotection or neurodegeneration.

## 1. Introduction

Polyamines (PAs) are organic polycations found ubiquitously in organisms, and, in mammals, they are mainly represented by putrescine (Put), spermidine (Spd), spermine (Spm) and their acetylated forms. In mammalian cells, PAs are involved in cell proliferation, differentiation, apoptosis, the synthesis of proteins and nucleic acids, the regulation of ion channel activity and the protection from oxidative injury [1,2,3]. Animals devoid of PA biosynthesis do not survive the early stages of embryonic development; notably, the supplementation of agmatine, a compound that belongs to the PA family and may serve as a precursor for Put (although exerting largely separate functions in mammal tissues), may be sufficient to rescue PA biosynthesis when the biosynthesis of Put is blocked (suggesting that the agmatine pathway is fully developed only later in life; for a discussion, see [4]). In fact, agmatine is a promising candidate for the treatment of several disorders, including neurodegenerative diseases, and highlights key roles for PAs in central nervous system (CNS) disorders [5].The cellular content of PAs is tightly regulated [1,2,3]; their biosynthesis is catalyzed by different enzymes including S-adenosylmethionine decarboxylase (AdoMetDC), ornithine decarboxylase (ODC), spermine synthase (SMS) and spermidine synthase (SRM) [6], while the enzymes N1-acetylpolyamine oxidase (PAOX), spermidine/spermine N1-acetyltransferase (SAT1) and spermine oxidase (SMOX) are involved in the PA catabolism (Figure 1) [7,8,9,10].

## 2. Polyamines in the Brain

Polyamines present a unique biochemistry in the brain, being primarily synthetized in neurons, while glial and microglial cells are mainly involved in their uptake and release. Furthermore, glial and microglial cells convert and oxidize PAs and release, besides hydrogen peroxide (H_2_O_2_), 3-aminopropanal (3-AP) and 3-acetoamidopropanal, hypusine, putreanine and gamma-aminobutyric acid (GABA) as end-products as well [4,11]. The major pathway of PAs to enter the brain was found through transporters on the astrocyte endfeet enwrapping blood vessels at the glial–blood interface [12], turning the focus on the mechanisms of uptake and replenishment in the case of an age-dependent decrease in PA synthesis. The most relevant PA transport system includes large pores such as connexins and pannexin hemichannels and transporters such as polyspecific organic cation transporters (OCTs), including solute carrier (SLC) 22A1-3 [4]. Such systems also transport monoamines such as dopamine or levodopa (l-dopa) [13] and may function in reverse mode releasing PAs, therefore regulating the extracellular PA levels and neuronal activity [12]. Another important transporter for PAs is the vesicular transporter SLC18B1, present in both neurons and astrocytes [14], which was found to be involved in regulating the PA content, the function of the GABA and glutamatergic systems, memory, synaptic function and plasticity [15,16]. Recently, the P-type ATPase transporter ATP13A3 has also been demonstrated to be involved in PA mammalian import [17,18]. In the adult brain, neurons are capable of synthetizing PAs [4,19,20], and glial cells but not neurons accumulate PAs [4,21,22,23,24]. However, the work of Masuko et al. [25] reported that PAs, especially Spm, are accumulated in synaptic vesicles and released by depolarization.

The fact that PAs such as Spd and Spm are taken up and accumulated by glial cells has led to a major focus on astrocytes. Notably, astrocytes, regardless of the true glia-to-neuron ratio and whether they outnumber neurons in the human brain or not [26,27], are no longer the “unacknowledged partner” [28] but are now fully “acknowledged” partners of neurons in CNS [29,30]. Two main functional consequences of PAs being stored in astrocytes can be envisaged: (a) PAs released from glia regulate the function of receptors and channels [31,32] in glia and neurons; (b) PAs stored in astrocytes regulate their own glial inward rectifier K^+^ (Kir)4.1 channels [33,34], connexin-43 (Cx43) channels [35,36] and GluA2-lacking alpha-amino-3-hydroxy-5-methyl-4-isoxazole-propionic acid (AMPA) receptors [37]. In fact, PAs released from astrocytes can affect both neurons and astrocytes while intracellular PAs regulate glial function. Notably, the astrocytic PA-sensitive receptors and channels, as well as the Put-to-GABA conversion [38], play roles in CNS diseases related to PAs [31,39]. On the other hand, astrocytes, by accumulating PAs in CNS [14,22,23], are involved in the protection against disease(s) and may be relevant to the Spd therapeutic potential [40,41,42,43,44,45,46].

Increasing evidence indicates that the dysregulation of the PA system is involved in neurodegeneration in different CNS pathological conditions. In the Snyder–Robinson syndrome, a mutation of the *SMS* gene, leading to a reduction of Spm and an accumulation of Spd, is responsible for a complex syndrome with intellectual disability, movement disorders and seizures [47]. Neurological abnormalities have been also reported in rodent models of the altered synthesis and catabolism of PAs [10,48]. Altered cellular levels of PAs and PA dyshomeostasis have been implicated in numerous brain diseases, including mental disorders, epilepsy [49], Alzheimer’s disease (AD) [50,51], Parkinson’s disease (PD) [52], traumatic brain injury [53] and in the pathogenesis of ischemic brain damage [54,55] and neurovascular damage in the retina [56,57]. In most of these conditions, dysregulation of the enzyme SMOX was reported. SMOX activity was also found to be increased in sera from schizophrenia patients, suggesting a key role of SMOX in the pathology [58]. Interestingly, in patients with neurocognitive impairment, SMOX over-expression caused by *SMG9* Nonsense Mediated MRNA Decay Factor (*SMG9*) loss-of-function, a gene key regulator of nonsense-mediated decay, was associated with intellectual disability [59].

## 3. Polyamine Catabolism: The Enzyme SMOX

Spermine oxidase catalyzes the conversions of Spm to Spd with the production of H_2_O_2_ and 3-AP [60,61,62] (Figure 2).

Spermine oxidase is a highly inducible enzyme [57] which is expressed mainly in the brain and skeletal muscle but also in the kidney, pancreas, bone marrow, lung, heart, intestine and spleen [10,63], where it regulates the Spm/Spd ratio to balance the cellular PA content, while SMOX dysregulation can alter the PA homeostasis [10,64]. The SMOX substrate Spm plays important functions in brain, since intracellular Spm acts as a neuromodulator responsible for the rectification of strong inward rectifier Kir channels [25,65,66] of AMPA and of kainate Ca^2+^-permeable receptors [67,68]. On the other hand, extracellular Spm can affect the function of *N*-methyl-D-aspartate (NMDA) glutamate receptors [25,65,66]. Additionally, Spd, the oxidation product of SMOX, functions as a neuromodulator, even though it is less potent than Spm [65]. New roles for Spd are increasingly investigated, such as its potential in protecting organisms from age-induced memory impairment through an autophagy-dependent homeostatic regulation at synapses [40] or in enhancing cerebral mitochondrial function and cognition in aging [42]. In fact, Spd has been reported to induce autophagy in model systems including rodent tissues and cultured human cells [43,69,70]. In *D. melanogaster*, autophagy seemed crucial for Spd protection against the presynaptic active zone changes during aging [41]. SMOX activity, in addition to keeping PA cellular content balanced, can alter the cellular redox homeostasis by producing H_2_O_2_, an endogenous reactive oxygen species (ROS). Although H_2_O_2_ plays roles in physiological brain function, excessive H_2_O_2_ production can result in learning and memory impairment [71]. Moreover, SMOX activity may be responsible for secondary tissue damage due to the generation of 3-AP, which spontaneously converts into acrolein [9,72], further inducing inflammation and apoptotic cell death in an injured brain [72].

Remarkably, growing evidence indicates a link between PA dyshomeostasis and neurodegenerative diseases. Polyamines are pivotal players in signalling responses to various environmental stimuli, which are involved in various aspects of the cellular metabolism, the maintenance of antioxidant capacity and osmotic regulation [1]. It is becoming clear that the PA role may shift from positive to negative in disease development, contributing to the shift from healthy to pathological conditions. The regulation of the PA system entails the balance of PA neuroprotective effects (in healthy conditions) versus detrimental effects of PA derivatives produced during oxidative stress and enduring stimuli. As a matter of fact, native PAs are ROS scavengers [73] and can play neuroprotectant roles, functioning as adaptive mechanisms maintaining homeostasis in CNS. Notably, the neuroprotectant effects of PAs focus on the roles of astrocyte–neuron crosstalk in maintaining healthy neuron–astrocyte network function. As already pointed out, while few adult neurons are synthesizing PAs but not holding PAs, astrocytes do not synthesize but collect PAs. Notably, the PA content declines during aging [74], and the loss of PA homeostatic mechanisms may be relevant to AD, PD and other age-related diseases. Indeed, PA loss correlates with the development of CNS disorders, and PA restoration has a rescuing effect [40]. The activation of the PA pathway and PA oxidation via SMOX or other oxidases, by generating ROS and by stimulating the antioxidant defence cell response (e.g., through the nuclear factor erythroid 2-related factor 2 [75]), may play both regulating and pathological roles, primarily in neurons and may shift the PA response from the protective, adaptive response-maintaining homeostasis towards a maladaptive detrimental mechanism. In fact, the increased PA levels following a short-term stimulus may have a beneficial role, while enduring stimuli such as repetitive brain trauma, cerebral arteriosclerosis-associated ischemia and metabolic stress lead to aberrant PA metabolism and, eventually, if it becomes maladaptive, to a deviant “PA stress response”, initiating the vicious cycle of neurodegeneration [76]. The continuous induction of the PA pathway is followed by arginine brain deprivation, the extensive catabolic oxidation of PAs, ROS generation and the induction of oxidative stress [76]. In neurodegenerative diseases, augmented PA metabolism results in the generation of H_2_O_2_ and reactive aldehydes including acrolein, which participate in the death of compromised tissue [77]. Indeed, while SMOX activity in the healthy brain was found in some neurons, SMOX upregulation and overexpression was found in both PD [78] and AD [51] and in some CNS diseases [79]. SMOX induction has been reported in diabetic retinopathy [56] as well as in brain ischemia [54]; in these pathological conditions, the upregulation of SMOX seems to be responsible for neuron damage (see Figure 3).

### 3.1. Alzheimer’s Disease

Starting from a pioneering observation of PA-altered levels in the autoptic brain of AD patients, where Spd was increased and Spm was reduced at the cortex level, it was suggested that abnormal PA activity may be involved in the neurodegenerative processes occurring in the brain of AD patients [50]. Polyamines dysregulation was confirmed in AD brains, and altered levels of PA metabolic enzyme transcripts (including an increase in SMOX mRNA) were suggested to promote tau neuropathology and induce cognitive and affective impairments [51]. PAs were proposed to trigger neurodegeneration in AD by condensing hyperphosphorylated tau [80]. While the AD-associated PA response may be envisioned as an integrated part of the conserved adaptive mechanism, the prolonged induction of PAs possesses a limited efficacy in coping with gradual oxidative stress and may have detrimental effects due to toxicity issues. The continuous induction of the PA pathway with the extensive catabolic oxidation of PAs, ROS generation and the induction of oxidative stress may aggravate AD [76]. Consistently, acrolein adducts are present in dystrophic neurites surrounding senile plaques [81], acrolein levels are significantly increased in AD patients’ hippocampi [82] and acrolein’s role in the AD pathogenesis has been suggested [76,83]. In fact, AD offers an example of how PAs are “wonderful machinery” in the interplay between neuroprotective and detrimental effects, which can drive the development of disease. One of the key elements of AD is the accumulation of amyloid beta (Aβ). Amyloids are strongly charged anions, and PAs are cations, so PAs, which can function as scavengers of ROS [73], may be bound by Aβ and neutralized or inactivated. Supplemental PA treatment is neuroprotective [40,41,69,84,85,86], while the oxidation of PAs can cause neurodegeneration ([87] and the current review). Therefore, glial cells, which are donors of PAs (unless Aβ or other acid proteins can buffer PAs), seem to be key players in the shift from synaptic function to dysfunction. In accordance, the dysregulation of astrocyte–neuron communication is considered to play major roles in neuron dysfunction in AD [88]; reactive astrocytes have been suggested to be a “double-edged sword” in AD, exerting biphasic effects—beneficial or detrimental, depending on multiple factors [89]. PAs may be one of these factors. As already mentioned, PA levels decrease with aging [74]; the condition may be dramatically different in young and aging brains, as SMOX activity is different [60], and recent evidence indicates that developing astrocytes can synthetize PAs, while adult astrocytes do not [46].

### 3.2. Parkinson’s Disease

In PD patients, the Spm/Spd ratio in blood was significantly decreased, indicating the Spm conversion from Spd and a decrease in brain Spm; the Spm/Spd ratio enabled the discernment between PD patients and healthy controls [90]. Interestingly, Spm was reported to prevent manganese-induced toxicity in dopaminergic neurons [91], while SMOX, activating Spm breakdown and leading to the excessive formation of toxic aldehydes (such as acrolein), H_2_O_2_ and ROS were found to be up-regulated in PD [92]. The role of PAs as gliotransmitters and regulators of neural function was suggested to be involved in triggering neuron oxidative stress and gliosis in PD [92]. Due to the central role of PAs in cell functionality, it is not surprising that PA dysregulation destabilizes neuronal function. A growing body of evidence indicates that the cation-transporting ATPase 13A2 (ATP13A2) and PAs play a key role in the endo-lysosomal system and mitochondrial function, which are at the heart of neurodegenerative diseases. Several reports directly support the importance of PAs homeostasis, beyond ATP13A2, in neurodegeneration, particularly PD [93]. ATP13A2 (also known as PARK9) is a lysosomal PA exporter with the highest affinity for Spm that promotes cellular PA uptake via endocytosis and PA transport into the cytosol [93]. Notably, loss-of-function mutations in the *ATP13A2* gene seem causally linked to neurodegenerative diseases, including Kufor-Rakeb syndrome, a rare form of inherited juvenile-early onset Parkinson’s disease [93,94,95,96,97,98,99,100,101], the while enhancement of ATP13A2 function has been proposed as a neuroprotective therapeutic strategy in Parkinson’s disease [100,102,103,104]. ATP13A2-associated disorders are hallmarked by mitochondrial and lysosomal abnormalities; in fact, ATP13A2 might exert a dual protective effect by preventing the lysosomal accumulation of PAs and by increasing the PAs’ cytosolic levels. The contribution to the intracellular pool of PAs may mediate the protective effect of ATP13A2 on mitochondrial toxins and heavy metals, as PAs are well-established ROS scavengers [73] and heavy metal scavengers and play essential roles in mitochondrial functionality and autophagy regulation [44,93].

### 3.3. Diabetic Retinopathy and Retinal Pathologies

The neurotoxicity of the oxidative products of PA degradation such as H_2_O_2_, acrolein and aminopropanal led to the “aldehyde load” hypothesis for neurodegenerative diseases [77,105]. Indeed, the inhibition of PA oxidases could prevent the NMDA-induced retinal neurodegeneration promoting cell survival, therefore providing a novel therapeutic target for retinal neurodegenerative disease conditions [106]. As a matter of fact, the evidence points to roles for trauma, oxidative stress and PA metabolism alteration in optic nerve injury, glaucoma or prematurity retinopathy, also suggesting the regulation of PA metabolism as a neuroprotection strategy [85,87,107]. Among PA oxidases, SMOX was reported to be involved in causing neurovascular damage in the retina [56] and was proposed as a therapeutic target for neurodegeneration in diabetic retinopathy [57]. Notably, the increased expression of SMOX was reported in the retina in response to hyperoxia-induced neuronal damage in retinopathy of prematurity [87] and excitotoxicity-induced retinal neurodegeneration [106]. The molecular mechanisms underlying neurodegeneration in the diabetic retina include the glutamate excitotoxicity and oxidative stress [108]. Mitochondria are one of the major targets of oxidative insults, and oxidative damage-mediated mitochondrial dysfunction is a major mechanism for neuronal damage in neurodegenerative diseases [109,110]. The proposed mechanism for SMOX-induced neuronal damage and dysfunction in diabetic retinopathy includes acrolein production, the depletion of antioxidants and mitochondrial dysfunction [57].

### 3.4. HIV-Associated Dementia

The enzyme SMOX is also responsible for the chronic oxidative stress occurring in the brain tissues of human immunodeficiency virus (HIV)-infected patients, leading to the pathogenesis of HIV-associated dementia [111]. In human neuroblastoma cell line HIV, the HIV-1 transactivator of transcription (Tat) elicits SMOX enzymatic activity upregulation through NMDA receptor triggering, thus increasing ROS generation, which in turn causes cell death [111]. Furthermore, ROS produced by SMOX can stimulate the antioxidant defence cell response through the nuclear translocation of Nrf2 (nuclear factor erythroid 2-related factor 2), which induces the expression of the oxidative stress-responsive genes [75]. These studies highlight that the NMDA/SMOX/Nrf2 pathway could be an important target for the protection against HIV-associated neurodegeneration [75].

## 4. SMOX as a Therapeutic Target to Treat Neurodegenerative Diseases

The identification of the “deviant PA stress response” in neurodegeneration has led to the hypothesis that the manipulation of PA catabolism is a realistic target for therapeutic or preventative intervention. To inhibit PA catabolism and, consequently, the aberrant production of ROS and acrolein, some inhibitors of PA oxidases have been designed, among which the most studied is N1, N4-bis(2,3-butadienyl)-1,4-butanediamine (MDL 72527), an irreversible competitive inhibitor [112]. MDL 72527 specifically inhibits SMOX and PAOX with similar inhibition constant (*Ki*)—21 and 63 μM, respectively—without affecting mono or di- amine oxidases [113]. MDL 72527 has been used in a wide range of experimental settings of neurodegeneration models. For example, the increase in the expression of SMOX during retinal excitotoxicity was associated with the degeneration of neurons, while MDL 72527 treatment improved neuronal survival and reduced neuroinflammation and microglial activation [106,114,115]. In a diabetic mouse model, treatment with MDL 72527 is able to ameliorate the diabetic retinopathy, restoring the survival of retinal ganglion cells as well as the structure and function of the retina [56].

The blockade of PA oxidation using MDL 72527 was found to be protective after traumatic brain injury [116]. Moreover, the use of N1-tridecyl-1,4-diaminobutane (C13-4), another competitive inhibitor of PAOX and SMOX, has also been demonstrated to be successful in the treatment of ischemic stroke to reduce the volume of brain infarction [117].

Kainate is a widely used inducer of excitotoxicity, and the co-treatment with MDL 72527 led to neuronal protection and attenuated lipid peroxidation, cytosolic cytochrome C release and glial cell activation in the hippocampus [118].

In a cellular model of HIV-associated neurocognitive disorders (HANDs), MDL 72527 restores cell viability and blunts Tat-induced ROS production [111]. In the same study, another inhibitor of PA oxidases, chlorhexidine, was also used, with comparable efficacy. Chlorhexidine acts as a strong—even stronger than MDL 72527—and competitive inhibitor of SMOX and PAOX, with a *Ki* of 0.55 and 0.1 μM, respectively, making it a promising lead compound to develop new selective inhibitors of PA oxidases [119].

## 5. A Genetic Model of SMOX Overexpression: The SMOX Overexpressing Mouse

The molecular mechanisms that might link the dysregulation of PAs with neuron damage have been investigated in a genetic model of chronic SMOX overexpression. A mouse model overexpressing *SMOX* in cerebrocortical neurons (Dach-SMOX) has been engineered [60,120]. Neuron loss and reactive astrocytosis are the main effects of the chronic activation of Spm catabolism in the neuronal cortex of Dach-SMOX mice [121,122,123]. This mouse model has not only been revealed to be a chronic model of excitotoxic/oxidative injury, showing both neuron damage and reactive astrocytosis [121,122,123], but also exhibits an increased susceptibility to epileptic seizures [60].

## 6. Neuron Damage in the SMOX Overexpressing Mouse

The Dach-SMOX cerebral cortex showed pronounced damage with significant neuron loss (reduction in the number of Neuronal Nuclei (NeuN) positive cells) [60,121] and neuron dysfunction (increase in the number of neurons with cytoplasmic condensation and nuclear basophilia in response to in vivo kainate treatment) [60]. Accordingly, a reduction in synaptophysin-positive particles indicated damage of the nerve terminals [121]. On the other hand, the nerve terminals, although maintaining a glutamate-releasing response to the activation of the AMPA receptor [121], displayed a reduced expression of both the GluA1 subunit and the GluA1 subunit phosphorylated at the serine 831 (Ser831) in the AMPA receptor [122]. The nerve terminal dysfunction in the Dach-SMOX cortex was confirmed by the impaired control of the Ca^2+^ signal in response to AMPA receptor activation [37], as well as by the catalase depletion being a sign of impairment in the antioxidant defence [37].

As a matter of fact, both chronic oxidative stress and glutamate transmission derangement in neuronal-glial networks might impair the neuron defence, contributing to vulnerability to oxidative and excitotoxic damage. Moreover, neuron dysfunction might impair the signalling from neurons to astrocytes at the synaptic level [124], in turn affecting the communication between astrocytes and neurons, and this signalling might involve PA. Reduced Spm release from neurons to astrocytes and reduced astrocytic Spm levels would impair the astrocyte Spm replenishment of neurons [12,37,125], causing trophic signalling deprivation. Both oxidative and excitotoxic chronical stress are likely to be related to neuron dysfunction and loss, as outlined below.

### 6.1. Chronic Oxidative Stress

Oxidative stress is recognized to play a role in different brain disorders [126,127]. The CNS, rich in unsaturated fatty acids, is therefore highly susceptible to oxidants such as ROS, particularly H_2_O_2_, that trigger lipid peroxidation, leading to neuron death [126]. The main enzymatic defence against ROS is played by the antioxidant enzymes SOD and catalase, which degrade superoxide radicals and H_2_O_2_, respectively, and by the non-enzymatic scavengers metallothioneins (MTs), which are up-regulated in animal models of neurodegenerative diseases [128,129,130]. In the CNS, MT-1 and MT-2 are largely expressed in astrocytes and play a neuroprotective role against heavy metals and oxidative stress [131], reducing the activation and recruitment of monocytes/macrophages and T cells and the activation of microglia [132]. MT-3 is abundant in neurons, where it seems to act in neuronal Zn^2+^ homeostasis [133]. In the brain cortex of Dach-SMOX mice, SMOX overexpression enhanced the production of H_2_O_2_ [60]; in accordance with the consequent chronic oxidative stress, the marker for oxidative DNA damage, 8-Oxo-2′-deoxyguanosine (8-oxo-dG) [134], was highly increased [122]. Consistent with chronic oxidative stress, Nrf2, a cellular defence against oxidative insults [135], was activated [122], and SOD and catalase activities and *MT* genes expression were increased in the cerebral cortex of Dach-SMOX mice [121]. Furthermore, an increased number of microglial cells was observed [60]. A deeper analysis (by assessing the morphology and function of purified nerve terminals and of astrocyte processes prepared from the cerebral cortex of SMOX-overexpressing mice) revealed the stimulation of catalase activity in astrocyte processes [37], while a reduction in catalase activity was found in the nerve terminals [37]. Such a reduction in the catalase activity in the nerve terminals appears to be consistent with neuron damage; notably, the deficiency or malfunctioning of catalase was found to be associated with neuron damage in neurodegenerative disorders such as AD and PD [136]. All these findings converge to indicate that, in the Dach-SMOX cerebral cortex, neurons and astrocytes undergo chronic oxidative stress. The chronic activation of defence mechanisms could maintain an oxidants/antioxidants balance, but, as a consequence of even small additive oxidative insults, the antioxidant system could be overwhelmed, resulting in accumulated cell damage in time [137,138]. This chronic condition is different from the acute generation of high amounts of the Spm oxidation products H_2_O_2_, 3-AP and acrolein in the cerebral ischemia (see the “aldehyde load” hypothesis for cerebral ischemia) [105], albeit both are responsible for neurotoxicity and cytotoxicity [139,140]. The continuous stimulation of PA catabolism with the chronic imbalance of radical homeostasis seems to be better related to the maladaptive “PA stress response” [141], which can aggravate chronic pathological CNS conditions involving mechanisms of oxidative activation and neurodegeneration, such as AD [76]. Notably, in neurodegenerative diseases, imbalance in the brain oxidants/antioxidants balance was described, originating from increased ROS production or the failure of the antioxidant defence [142].

### 6.2. Chronic Excitotoxic Stress

The activation of the excitotoxic mechanism, with the involvement of astrocytes, is a well-known common pathway for neuron damage in chronic and acute CNS disorders [143,144,145]. In the cerebral cortex of Dach-SMOX mice, reactive astrocytes can participate in raising the extracellular glutamate levels, directly contributing to chronic excitotoxicity. In fact, the reactive astrocyte processes expressed GluA2-lacking AMPA receptors, which allowed for the entry of Ca^2+^ and activated glutamate release in response to glutamate, therefore taking part in a positive feedback loop [37,121]. Moreover, the reactive astrocyte processes contributed to an increase in extracellular glutamate through the activation of the cystine-glutamate transporter xc^-^ transporter [122] and a likely impaired glutamate clearance from the synapse [146] due to a reduced expression of the astrocytic excitatory amino acid transporters EAATs [122]. Notably, the ability to buffer Ca^2+^ seems to be crucial for neuron susceptibility to excitotoxic insults [147]; the impaired buffering of Ca^2+^ in the nerve terminals [37], after the activation of the AMPA receptor, may contribute to neuron vulnerability to excitotoxic damage in Dach-SMOX mice.

Therefore, the detrimental effects of reactive astrocyte processes seem to be important determinants in neuron dysfunction and loss in Dach-SMOX mice, being involved in excitotoxic mechanism activation and possibly in a reduced supply of neuroprotective factors. Moreover, neuron dysfunction might impair the signalling from neurons to astrocytes at the synapse [124], in turn affecting the communication between astrocytes and neurons. The signalling could involve PA. A reduction in Spm release from the neurons to the astrocytes in Dach-SMOX mice might lead to reduced astrocytic Spm levels and to a reduced ability of astrocyte to replenish neurons with Spm [12,37,125] in a self-sustaining deprivation of trophic signalling. The findings in this chronic model of PA catabolism activation suggest that endogenous PAs play roles in maintaining neuron–astrocytes intercellular signalling. In fact, during the continuous induction of PA catabolism, neurons may undergo recurring astrocyte-dependent insults by potentially self-sustaining glutamate excitotoxic cascades and by potentially self-sustaining the deprivation of trophic/neuroprotective factors.

## 7. Reactive Astrocytes in the SMOX Overexpressing Mouse

Neuron–astrocyte intercellular communication is recognized as crucial for signal transmission and the regulation of brain function [148,149,150,151]. Astrocytes provide structural support at the synapses, energy substrates and neurotransmitter precursors to neurons, the buffering of K^+^ [152,153,154,155] and the regulation of extracellular glutamate by balancing the uptake through EAAT1 and EAAT2 and the release and uptake through the cystine-glutamate exchanger xc^-^ [156,157]. Furthermore, the fine perisynaptic astrocytic processes (PAPs) unsheathing synapses [158] can release glutamate in Ca^2+^-dependent vesicular or Ca^2+^-independent ways [29,30,121,122,159,160,161,162,163,164,165,166]. Astrocytes undergo remodelling, the so-called reactive astrocytosis, in response to brain injury and to neuroinflammation conditions [153,167,168]; reactive astrocytes may have both detrimental and neuroprotective actions [169]. Astrocytes in the Dach-SMOX cerebral cortex undergo reactive astrocytosis, as indicated by the increase in the number of astrocytes and by astrocyte hypertrophy and wide ramification [60,121]. Consistently, the relative abundance of astrocyte processes vs. nerve terminals was higher in Dach-SMOX mice, with increased Glial Fibrillary Acidic Protein (GFAP)-positive particles [121] and increased levels of the astroglial markers ezrin (a protein preferentially localized in PAPs [158], required for PAPs motility and the regulation of synapse coverage [170] and possibly participating in neuroprotective and neurotoxic activities of the reactive processes [158]) and vimentin (a potential marker for reactive astrocytes [168,169] relevant to the function of astrocyte and astrocyte processes in reactive astrocytosis) [37,171]. Reactive astrocytosis in the Dach-SMOX cerebral cortex is likely dependent on the overproduction of H_2_O_2_ and on neuron dysfunction. Indeed, oxidative stress and inflammation can promote reactive astrocytosis [172], reactive astrocytes in turn being able to generate ROS [173]. In addition, PAs released from neurons to glial cells [12,25,125] may be hypothesized to be trophic signals from synaptic neuronal activity crucial to maintaining healthy astrocytes [124]. Reactive astrocytes in Dach-SMOX mice may contribute to synapse dysfunction and to detrimental effects on neurons [37] via the following mechanisms: reduced Spm content, the expression of functional AMPA GluA2-lacking receptors, increased xc^-^ function and the reduced expression of EAAT1 and EAAT2.

### 7.1. Reduced Spm Content

The PA levels remained unchanged in the cerebrocortical nerve terminals of SMOX-over-expressing mice as compared to controls, despite SMOX overexpression in neurons, while the level of Spm was reduced in the astrocyte processes [37]. This indicates that PAs are synthesized in neurons, are released into the extracellular space and are then almost exclusively stored in glial cells [25], which in turn can release them to regulate neuronal synaptic activity [12,125]. The low level of Spm in astrocyte processes can be explained by hypothesizing that astrocytes secrete Spm [12,45,46,125] to replenish neurons and keep the concentration of Spm as well as the Spm/Spd balance constant within neurons. The findings, together with the absence of a change of the Spm content in the cerebral cortex of Dach-SMOX mice [60], are consistent with the tight regulation of PA homeostasis. In fact, Spm acts as an intracellular blocker of Ca^2+^-permeable AMPA and kainate receptors [67,68,174], and Kir4.1 channels [32,33,65] and reduced Spm content could take part in the dysfunction of astrocyte processes in Dach-SMOX mice, possibly contributing to a reversion of the transformation of reactive astrocytes from neuroprotective to detrimental.

### 7.2. Expression of Functional AMPA GluA2-Lacking Receptors

The expression of the subunit GluA1 of the AMPA glutamate receptor and of its phosphorylated form at Ser 831 site increased in the astrocyte processes of Dach-SMOX mice [122]. Notably, in Dach-SMOX mice, increased ROS production [60] and higher levels of PKC [123] were found, and an increase in the GluA1Ser831 and AMPA receptor localization at the plasma membrane were found to be ROS [175] and PKC activation-dependent [176]. AMPA receptor activation, completely ineffective in the processes from control mice [37,121], evoked Ca^2+^ influx [37] and glutamate release [121] in Dach-SMOX mice astrocyte processes. Low Spm levels in the processes fit in with the functioning of the Ca^2+^-permeable GluA2-lacking AMPA receptors [67,68,174]. The ability of the GluA2-lacking AMPA receptors to evoke Ca^2+^ influx and its coupling to vesicular glutamate release in the SMOX astrocyte processes appear to be of relevance for a better understanding of Ca^2+^ microdomains [177,178] in PAPs from reactive astrocytes.

### 7.3. Increased xc^-^ Function

The transporter xc^-^, expressed mainly on astrocytes [179,180], imports cystine and exports glutamate [122]. The expression of this transporter, directly regulated by Nrf2 [179,180], was increased in Dach-SMOX mice [122], and a greater glutamate-releasing response to extracellular cystine was observed in Dach-SMOX astrocyte processes as compared to controls [122]. The chronic oxidative stress in Dach-SMOX mice is likely to be responsible for the transporter’s increased expression and, consequently, for the higher glutamate release from astrocytes.

### 7.4. Reduced Expression of EAAT1 and EAAT2

The excitatory amino acid transporters EAAT-1 and EAAT-2 are mainly localized on the membrane of the astrocyte processes and are involved in the clearance of synaptic glutamate [146]. In Dach-SMOX mice, the impairment of EAATs expression in excitotoxic conditions was reported [122], probably as a consequence of oxidative stress [181,182]. The lower expression of EAAT-1 may be responsible for the impairment of the astrocytic clearance of glutamate from the synapse, therefore potentiating excitotoxicity.

Therefore, the processes of reactive astrocytes in Dach-SMOX mice may sustain a positive loop: the neuronal glutamate released could activate astrocytic AMPA receptors, evoking the further release of glutamate, which, together with increased glutamate release through the astrocytic xc^-^ exchange and reduced astrocytic glutamate uptake by EAATs, further increases extracellular glutamate, contributing to a chronic increase in neuronal excitability and excitotoxicity (Figure 4). Notably, the altered astrocyte function appeared to also be involved in the increased susceptibility to kainate seizures [121,183].

## 8. Conclusions

The main points to understand the linkage between PA dysregulation and neurodegeneration can be summarized as follow: neurons synthesize PAs, while astrocytes do not synthesize them but store them; PAs are collected by adult glial cells that can no longer be considered “unacknowledged partners of neurons” and provide a new avenue in the neuroscience of PAs; SMOX activity in the normal brain was found in some neurons but was overexpressed in some instances of CNS disease; PA catabolism, mainly via SMOX, is involved in the crosstalk between neurons and astrocytes in neurodegeneration.

Taken together, the evidence in the SMOX overexpressing mouse model shows that the chronic activation of PA catabolism and the consequent H_2_O_2_ overproduction in cortical neurons affect astrocytes and in turn neurons, underlining the requirement of neuron PA pathway regulation for neuron health and protection. The findings support a pivotal role for PAs in neuron–astrocyte cross-talk, with effects on neuroprotection [125], helping to elucidate mechanisms through which reactive astrocytes can affect the astrocyte–neuron communication, leading to neuron damage and neurodegeneration.

In addition, the SMOX-overexpressing mouse might help in shedding light on the mechanisms underlying epileptogenesis. The mouse recapitulates the conditions, namely, the excitotoxic mechanism [144,145,184], oxidative stress [185,186] and reactive astrocytosis [103,109,110,111], which have been linked to epileptogenesis [144,145,186,187,188]. Notably, in the cerebral cortex of the Dach-SMOX mouse model, all these conditions depend on the chronic activation of neuronal PA catabolism.

Furthermore, the findings in the SMOX-overexpressing mice appear to be of relevance to the shift from a neurocentric to a neuro-astrocentric view [189] of brain function.

## Figures and Tables

**Figure 1 biomedicines-10-01756-f001:**
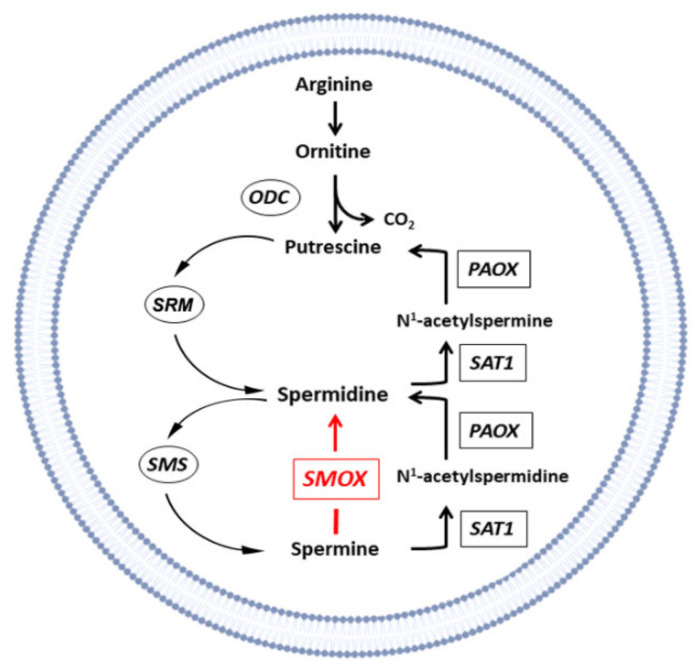
Enzymes involved in PA biosynthesis (encircled) and catabolism (boxed). ODC, ornithine decarboxylase enzyme; PAOX, N1-acetylpolyamine oxidase; SAT1, spermidine/spermine N1-acetyltransferase; SMS, spermine synthase; SRM, spermidine synthase. The enzyme spermine oxidase (SMOX), which is overexpressed in the Dach-SMOX mice model, is highlighted in red.

**Figure 2 biomedicines-10-01756-f002:**
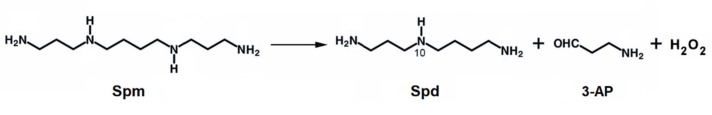
Spermine oxidase chemical reaction. Spermine (Spm) is oxidized to produce spermidine (Spd), 3-aminopropanal (3-AP) and hydrogen peroxide (H_2_O_2_).

**Figure 3 biomedicines-10-01756-f003:**
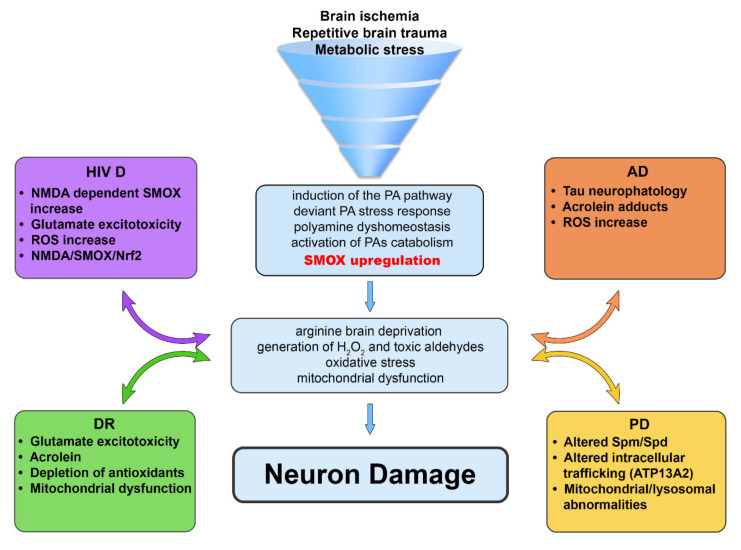
Schematic representation of the major mechanisms involved in neuronal damage resulting from polyamine dyshomeostasis in the central nervous system. Polyamine dyshomeostasis-dependent mechanisms that have been suggested to play pivotal roles in representative relevant diseases are also highlighted. AD, Alzheimer’s disease; DR, diabetic retinopathy; HIV D, HIV-associated dementia; PD, Parkinson’s disease.

**Figure 4 biomedicines-10-01756-f004:**
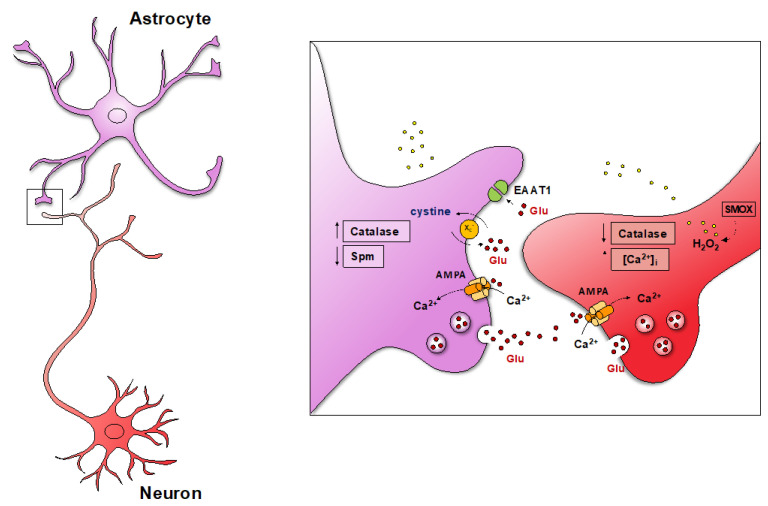
SMOX overexpression in neurons resulted in chronic oxidative and excitotoxic stress and in neuron loss. Schematic representation of the main mechanisms taking place at cerebrocortical glutamatergic synapses in the SMOX-overexpressing mouse model. NeuN positive cells were reduced [37,121], and a relative increase in the abundance of astrocyte processes and a decrease in nerve terminals (an increase in GFAP, ezrin and vimentin-positive cells vs. a reduction in synaptophysin and NeuN-positive cells) were found. SMOX overexpression in neurons leads to oxidative stress in neurons, increased by an ROS response in astrocytes and leading to the depletion of catalase (a reduction in the antioxidant defence in nerve terminals). A defective control of the AMPA-evoked intracellular Ca^2+^ response in the nerve terminals can exacerbate the reactive astrocytes-dependent excitotoxic mechanism activation. For further details, see the text. AMPA, alpha-amino-3-hydroxy-5-methyl-4-isoxazole-propionic acid receptor; Glu, glutamate; SMOX, spermine oxidase; Spm, spermine.

## Data Availability

Not Applicable.

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
