# Peer review of "The Involvement of Polyamines Catabolism in the Crosstalk between Neurons and Astrocytes in Neurodegeneration"

_biomedicines, 2022, doi:10.3390/biomedicines10071756_

Round 1

Reviewer 1 Report

TO AUTHORS:

The manuscript should be of high priority because  focusing on three important issues:

 (i)  polyamines are collected by adult glial cells that are "unacknowledged partners of neurons" and the data will give a new avenue in the neuroscience of PAs, (ii) few adult neurons and synaptic terminals are synthesizing PAs but not holding PAs while astrocytes do not synthesize but collect; (iii) SMOX activity in the normal brain was found in some neurons, but overexpressed in some CNS disease (ref. Bernstein et al., 2021, see below).  Therefore, PA oxidation via SMOX, PAO, DAO and MAO enzymes may play both regulating and pathological roles (Alzheimer’s disease (AD) primarily in neurons. Please, resubmit your manuscript covering important points of critics highlighted. 

Please note that the situation in young and adults, specifically aging brains may be dramatically different because not only SMOX activity is different but also recent evidence suggests that developing astroglial cells contain PA-synthesis while adult astrocytes do not. Catalytically active ODC in juvenal astrocytes was found, thus glial cells synthesize PAs and release them (Malpica-Nieves et al., 2020).

The analysis of the role of PAs in AD is a key point that is unequivocally posted in your review. Indeed, a robust amount of neurological literature ignores polyamines still.

I recommend authors include data and discussions from the following articles:

First, the idea that PAs are major players in many diseases is flourishing specifically in respect to study functional changes in CNS, but it is still an empty space in respect of (i) localization of SMOX, PAOs, DAOs and MAOs in CNS. Authors should cite the first SMOX study by Kauppinen and Alhonen (1995) showing that SMOX overexpression leads to glutamate toxicity as was later supported by your research, Pietropaoli et al. (2018). Please, cite:

Kauppinen RA, Alhonen LI. (1995) Transgenic animals as models in the study of the neurobiological role of polyamines. Prog Neurobiol. 47:545-563.

Also, in the study by Flayeh (1988), SMOX activity was found increased in sera from schizophrenia patients compared to non-psychotic controls which shows a key role of SMOX in pathology. Therefore, you need to  highlight these findings as a key in pathology in the brain linked to SMOX and cite it: 

Flayeh KA. (1988) Spermidine oxidase activity in serum of normal and schizophrenic subjects. Clin Chem. 34:401-403.

In this relation, please discuss and cite that the localization of spermine oxidase was found in neurons but not in astrocytes:

Bernstein HG, Keilhoff G, Laube G, Dobrowolny H, Steiner J. (2021) Polyamines and polyamine-metabolizing enzymes in schizophrenia: Current knowledge and concepts of therapy. World J Psychiatry. 11(12):1177-1190. doi: 10.5498/wjp.v11.i12.1177

Second, cite please recent articles showing the unique biochemistry of PAs in the brain and glial and microglial cell roles including PA accumulation, conversion and oxidation as well as the release of hypusine, putreanine and GABA (unique gliotransmitter) as end-products:

1.

Rieck J, Skatchkov SN, Derst C, Eaton MJ, Veh RW (2022) Unique Chemistry, Intake, and Metabolism of Polyamines in the Central Nervous System (CNS) and Its BodyBiomolecules. 12(4):501. doi: 10.3390/biom12040501.

2.

Chia TY, Zolp A, Miska J. (2022) Polyamine Immunometabolism: Central Regulators of Inflammation, Cancer and Autoimmunity. Cells. 11(5):896. doi: 10.3390/cells11050896

3.

Laube & Bernstein (2017) highlighted a key role of PAs in CNS syndromes, disorders and diseases where one of the most promising candidates is agmatine. With respect to major CNS disorders including depression and AZ, PD diseases, agmatine and its derivatives as well as metabolizing enzymes show great promise for the development of improved treatment of these common diseases. Therefore, the authors need to enlarge the scope of the controversy and a striking difference in results studying CNS diseases and PA relationship.

It should be cited:

Laube G; Bernstein HG; Agmatine: multifunctional arginine metabolite and magic bullet in clinical neuroscience? Biochem J. 2017 474(15):2619-2640. doi: 10.1042/BCJ20170007.

Third, the authors stated that polyamine stored in glial cells giving citation to Masuko et al., 2003. This is not the first finding, but much later than Veh & Laube, (1997); Biedermann et al., (1998); Bernstein et al., (1999) and Skatchkov et al., (2000) that demonstrated exclusive glial cells but not in neurons  capability to accumulate polyamines. Therefore,  authors need to highlight this fact, discuss and cite the following:

Laube G; Veh R.W.  (1997) Astrocytes, not neurons, show the most prominent staining for spermidine/spermine-like immunoreactivity in adult rat brain. Glia, 19: 171-9.

Biedermann, B., Skatchkov S.N.; Brunk I.; Bringmann A.; Pannicke T.; Bernstein H.G.; Faude F.; Germer A.; Veh R.W.; Reichenbach A. (1998), Spermine/spermidine is expressed by retinal glial (Muller) cells and controls distinct K+ channels of their membrane. Glia, 23: 209-20.

Skatchkov S.N.; Eaton M.J.; Krusek J.; Veh R.W.; Biedermann B.; Bringmann A.; Pannicke T.; Orkand R.K.; Reichenbach A. (2000) Spatial distribution of spermine/spermidine content and K(+)-current rectification in frog retinal glial (Muller) cells. Glia, 31: 84-90.

Fourth, there are two functional links of PAs stored in glial cells: (1) as a possible origin of the problem rising during disease development because PAs are released from glia and regulate receptors and channels (Nichols and Lee, 2018; Bowie, 2018) in glia and neurons; (2) while PAs stored in glial cells regulate own glial Kir4.1 channels (Kucheryavykh et al., 2007; 2008) and connexin-43 (Cx43) channels (Benedikt et al., 2012; Skatchkov et al., 2015). Therefore, when glial cells release PAs it first (i) affect neurons while normally (ii) PAs regulate glial function.

Therefore, the astrocytes outnumber neurons about ten times in the brainstem (Lent et al., 2012), carrying polyamines and expressing PA-sensitive channels, it is highly recommended to authors to note the role of astrocytic Kir4.1, TASK (2-pore domain channels), Cx43 channels and converting putrescine to GABA in astrocytes *Kovasc et al., 2022) that all play role in CNS diseases related to polyamines. When the articles are cited it is good support for your review and for the Journal:

Olsen M.L.; Khakh B.; Skatchkov S.N.; Zhou M.; Lee J.; Rouach N. (2015) New insights on astrocyte ion channels: Critical for homeostasis and neuron-glia signaling. J. Neuroscience,  35:13827-35.   

Bowie D. (2018) Polyamine-mediated channel block of ionotropic glutamate receptors and its regulation by auxiliary proteins. J Biol Chem, 293(48):18789-18802. doi: 10.1074/jbc.TM118.003794.

Fifth, the major pathway of monoamines and PAs to enter the brain was found in glial-blood interface it turns the focus on the mechanisms  of uptake and replenishment in case of age-dependent decrease of PA synthesis. The authors cites some PA transport system but not major, such as SLC18B1, SLC22A1, -2, -3 (OCT system)

For example, Inyushin et al. (2012) showed that L-dopa (the drug broadly used as anti-PD treatment) is (i) taken up from blood circulation to the brain via glial transporter and (ii) glial cells in the brain contain an enzyme, monoamine oxidase-B (MAO-B) that can convert monoamines and polyamines to toxic forms.

Inyushin, M.Y., Huertas, A., Kucheryavykh, Y.V., Kucheryavykh, L.Y., Tsydzik, V., Sanabria, P., Eaton, M.J., Skatchkov, S.N., Rojas, L.V. and Wessinger, W.D. (2012) L-DOPA Uptake in Astrocytic Endfeet Enwrapping Blood Vessels in Rat Brain. Parkinsons Dis-US., 2012, 321406-321414.

Sixth, the author’s attention should be paid specifically to the fact that during aging (related not only to AD, but neuropathy, stroke, Parkinson’s, and other age-related disorders and diseases) the polyamine content declines (Minois et al., 2011) while restoring of polyamines have rescuing effect (Bhukel et al., 2017). Aging and PA loss correlate with the development of many syndromes and CNS disorders as well.

Authors should cite:

Minois N.; Carmona-Gutierrez D.; Madeo F. Polyamines in aging and disease. Aging (Albany NY), 2011, 3, 716-32.

Seventh, since PAs, such as spermidine and spermine, are taken up by glial cells (see above) but glial cells do not synthesize spermidine by themselves (Krauss et al., 2006; 2007) it brings a major focus on glial cells that do not produce but accumulate PAs. Authors should cite;

 Krauss M.; Weiss T.; Langnaese K.; Richter K.; Kowski A.; Veh R.W.; Laube G. (2007) Cellular and subcellular rat brain spermidine synthase expression patterns suggest region-specific roles for polyamines, including cerebellar pre-synaptic function. J. Neurochem., 103: 679-93.

Therefore, because PA accumulation in CNS is by glial cells (Laube and Veh, 1997; Biedermann et al., 1998; Hiasa et al., 2014) that will determine the protection against disease(s) and the pharmacological treatment against the disease(s) since spermidine, for example, is rescuing many CNS function (Madeo et al., 2010; 2018; Gupta et al., 2016; Bhukel et al., 2017; Hofer et al., 2021; Malpica-Nieves et al., 2020; 2021).

Finally, this review will gain much more interest and citations if the authors include all recommendations above and discuss such wonderful machinery of PA neuroprotective effects versus malefic effects of PA-derivatives specifically produced during oxidative stress. Authors need to discuss the controversy of positive versus negative effects of polyamines in the development of the diseases.

It is clear that polyamines are involved in the pathogenesis and the oxygen stress and the oxidative PA pathway build ROS, while native PAs are rather scavengers of ROS and inflammatory molecules!

One of the key elements of AD is the accumulation of beta-amyloids. Since amyloids are strongly charged anions, but PAs are cations,  therefore polyamines may be bound  by beta-Amyloids and be neutralized, or inactivated. Therefore, taking into attention that supplemental polyamine treatment is neuroprotective (Noro et al., 2014; Bhukel et al., 2017; Sigrist et al., 2014; Gupta et al., 2013; Eizenberg et al., 2009)  while oxidation of polyamines causes neurodegeneration (Narayanan et al., 2014 and the current review) it should be strongly stated that glial cells are key players since are donors of PAs (unless beta-amyloids or other acid proteins can buffer PAs).  

SO, both, binding of PAs and oxidation of polyamines, are potentially lead to gliosis and this turns glial cells into unhealthy function and AD. The mechanisms of a switch to oxidation are not clear. PA-related gliosis is not due to “healthy” polyamines but can be triggered by PA-oxidation in AD by inflammation, trauma, infection etc.

Authors should cite important article pointing trauma and oxidative stress and role of polyamines:

Noro T.; Namekata K.; Kimura A.; Guo X.; Azuchi Y.; Harada C.; Nakano T.; Tsuneoka H.; Harada T. Spermidine promotes retinal ganglion cell survival and optic nerve regeneration in adult mice following optic nerve injury. Cell Death Dis. 2015, 6, e1720. doi: 10.1038/cddis.2015.93.

Kimura A.; Namekata K.; Guo X.; Noro T.; Harada C.; Harada T. Targeting Oxidative Stress for Treatment of Glaucoma and Optic Neuritis. Oxid Med Cell Longev. 2017, 2017:2817252. doi: 0.1155/2017/2817252

Narayanan S.P.; Xu Z.; Putluri N.; Sreekumar A.; Lemtalsi T.; Caldwell R.W.; Caldwell R.B. Arginase 2 deficiency reduces hyperoxia-mediated retinal neurodegeneration through the regulation of polyamine metabolism. Cell Death Dis. 2014, 5, e1075. doi: 10.1038/cddis.2014.23.

MINOR REMARKS

1.       The sentence is repeated twice in lines  200-203 and lines 264-267. It should be modified.

2.       The literature should be in proper and consequent order:

-The names should be cited in uniform mode. Please check  References # 13, 41, 47, 55, 74, 96, 102, 121, Parpura Vladimir,- should be written Parpura V.;  Verkhratsky Alexey should be written Verkhratsky A., and so on. Please, check all list of REFs.  Certainly, in eight references names need to be corrected.

3.        The page numbers are absent in REFs 2, 13, 18, 40, 47, 61, 96, 100, 111, 129, 142, 146. Please, when using EndNote program always check text after.

4.       In Reference # 74, there is no journal name.

5.       The literature should be supplemented/enlarged by other key articles in the area suggested by the Reviewer.

I recommend your manuscript for Minor Revision after answering some rigorous comments. Please, resubmit your manuscript answering minor but important points highlighted. Your manuscript is of high priority that is focusing on a new avenue in the neuroscience of polyamines and diseases.

Reviewer 2 Report

The review article entitled” Neurodegeneration in polyamine catabolism activation” discusses major mechanisms associated with neurodegeneration occurring in response to polyamine catabolism. The research group by Dr.Cervelli is well known in the field and has been publishing in the area of polyamine metabolism for several years. The current review summarizes the literature on the impact of polyamine catabolism, with a focus on SMOX function on neurodegenerative diseases.

Comments

1.     The introduction summarizes major pathways of polyamine metabolism and the dysregulation of PA system in neurodegenerative diseases. While the authors covered several CNS disorders, ocular diseases are not included. It would be beneficial to the general audience to include it in the introduction.

2.       Under section 2, including a schematic representation of the major mechanisms involved in neuronal damage resulting from polyamine metabolism is suggested. Fig 3 (section 5) is more specific on the mechanisms related to SMOX overexpression.

3.       Page 2, line 72 states “Spermine oxidase is a highly inducible enzyme [22], which is expressed mainly in brain and skeletal muscle,..”. However, SMOX is also present in the kidney, retina, intestine, etc. This section needs to be revised.

4.       It would be easier for the readers if Section 2 could be revised to include subsections on various pathologies already discussed (Alzheimer’s, Parkinson’s, Retinal diseases, Glioma, etc).

5.       A table with abbreviations can be added, and expanding the abbreviation when the first time used is suggested.

6.       A section discussing how SMOX can be targeted to treat neurodegenerative diseases is recommended to include.

7.       Another suggestion is to revise the title to be more clear and more specific.

Reviewer 3 Report

The authors have discussed a study topic for polyamine. The article is quite interesting, and I do agree that polyamine catabolism plays a vital role in neurodegeneration, astrocyte regulation, oxidative stress regulation, and so on. However, it is full of writing without sufficient illustrations, so readers can easily lose attention. I would recommend you to write as simply as possible with some tables and figures. Overall, the merit of the manuscript is good and follows a logical way. Still, there are some major modifications that the authors need to be addressed in their revised version.

1.     I believe the title should be changed as the story is all about polyamine with neurodegeneration, oxidative stress, astrocytes, etc. So, it would be more fit if the authors bring the term Crosstalk among polyamines, oxidative stress, astrocytes, or something like this.

2.     Could you please make a tabular format or graphical format stating the regulatory effects, significance, or importance with neuro disorders or disease names or OS regulation or astrocytic activation, etc? That would be easier to understand the roles and significance of PA.

Round 2

Reviewer 3 Report

Thanks for providing the corrections and now it is acceptable to me.

Congratulations.